# Residents transitioning between hospital and care homes: protocol for codesigning a systems-level response to safety issues (SafeST study)

Jason Scott [1], Katie Brittain,[2] Kate Byrnes,[1] Pam Dawson,[3] Stephanie Mulrine,[1] Michele Spencer,[4] Justin Waring,[5] Lesley Young-Murphy[6]

[1]Faculty of Health and Life Sciences, Northumbria University, Newcastle upon Tyne, UK
[2]Population Health Sciences Institute, Newcastle University, Newcastle upon Tyne, UK
[3]School of Sport, Health and Wellbeing, Plymouth Marjon University, Plymouth, UK
[4]North Tyneside Community and Health Care Forum, North Shields, UK
[5]Health Services Management Centre, University of Birmingham, Birmingham, UK
[6]NHS North Tyneside Clinical Commissioning Group, North Shields, UK

**Correspondence to**
Dr Jason Scott;
jason.scott@northumbria.ac.uk

## ABSTRACT

**Introduction** The aim of this study is to develop a better understanding of incident reporting in relation to transitions in care between hospital and care home, and to codesign a systems-level response to safety issues for patients transitioning between hospital and care home.

**Methods and analysis** Two workstreams (W) will run in parallel. W1 will aim to develop a taxonomy of incident reporting in care homes, underpinned by structured interviews (N=150) with care home representatives, scoping review of care home incident reporting systems, and a review of incident reporting policy related to care homes. The taxonomy will be developed using a standardised approach to taxonomy development. W2 will be structured in three phases (P). P1a will consist of ≤40 interviews with care home staff to develop a better understanding of their specific internal systems for reporting incidents, and P1b will include ≤30 interviews with others involved in transitions between hospital and care home. P1a and P1b will also examine the impact of the SARS-CoV-2 pandemic on safe transitions. P2 will consist of a retrospective documentary analysis of care home data relating to resident transitions, with data size and sampling determined based on data sources identified in P1a. A validated data extraction form will be adapted before use. P3 will consist of four validation and codesign workshops to develop a service specification using National Health Service Improvement's service specification framework, which will then be mapped against existing systems and recommendations produced. Framework analysis informed by the heuristic of systemic risk factors will be the primary mode of analysis, with content analysis used for analysing incident reports.

**Ethics and dissemination** The study has received university ethical approval and Health Research Authority approval. Findings will be disseminated to commissioners, providers and regulators who will be able to use the codesigned service specification to improve integrated care.

## INTRODUCTION

With the release of An Organisation with a Memory[1] and theoretical work on patient safety,[2] a movement for patient safety began which emphasised a systems approach to

---

### Strengths and limitations of this study

► A key strength of the Safe System Transitions study is it will be the first research examining how safety incidents are reported across the care home sector in relation to transitions in care using data generated within the care home sector.

► The study will capture qualitative insight and reflection on patient safety from those involved with and responsible for patient transitions across healthcare and care home settings during the COVID-19 pandemic.

► The development of a taxonomy of incident reporting systems within the care home sector will provide the foundations required for implementing changes that are required for making improvements to patient safety.

► Codesigning a service specification will define the service standards expected from organisations involved in patient transitions, and help to begin addressing the under-reporting of incidents.

► A limitation is that workstream 2 is focused on only two regions of England, though application of the developed taxonomy will inform whether the findings will be applicable across the wider health and social care sector.

---

safety. As a result, in the UK, the National Patient Safety Agency was launched in 2001, which in 2003 established the National Reporting and Learning System (NRLS). The NRLS is a central safety management system containing all patient safety incident reports from National Health Service (NHS) organisations, though it is now situated within NHS England and Improvement[3] and is currently being replaced with the Learn from patient safety events service. Reporting systems such as the NRLS have been important for improving patient safety internationally, particularly for incident types that require larger (eg, national) solutions, such as medication errors.[4] However, in England, there is

no such system that is able to consistently capture safety incident reports for all levels of harm or for near misses that occur when a patient transitions out of hospital and into a care home setting.

The safety of a transition between hospital and care home is less well understood than other settings such as emergency admission[5 6] or during the hospital stay.[7 8] Transitions between hospital and care home are also particularly high in risk, with a third of transitions resulting in adverse events.[9] Common challenges include communication failures,[10] medication errors[11] and incorrect documentation.[12] From an organisational governance perspective, identifying safety incidents that relate to transitions in care is especially difficult; care home and hospital organisations have different priorities,[13] health and social care sectors use different definitions of safety,[14] and efforts to involve patients directly have had mixed results.[15 16] Furthermore, the patient would be outside of the hospital's responsibility when the incident would be identified, and therefore it would be unlikely that a hospital staff member contacted by a care home would proceed to report a safety incident. Consequently, integrated care between the health and social care sectors is lacking in relation to patient safety, and opportunities for organisational, cross-sector learning are likely being missed.[14]

The COVID-19 pandemic, caused by the SARS-CoV-2, has had a significant impact on care homes within England due in part to a poor policy response.[17] The pandemic has also placed a specific focus on the safety and appropriateness of transitions between hospital and care home, with unsafe hospital discharge into care homes being identified as a cause of anxiety among care home staff.[18]

### Aims and objectives

This study aims to develop a better understanding of incident reporting in relation to transitions in care between hospital and care home, and to codesign a systems-level response to safety issues for patients transitioning between hospital and care home. To meet these aims, the study has the following seven objectives:

1. Investigate, using desk-based approaches, what policies exist for incident reporting, the technology used to incident report, and the types of data captured within incident reports.
2. Develop a taxonomy of approaches to incident reporting within care homes.
3. Identify, using qualitative methods, the sociotechnical and cultural determinants of incident reporting in care homes in relation to patient transitions, including how care homes report safety incidents and how decisions are made to report or not.
4. Conduct a retrospective documentary content analysis of incident reports (and similar systems) relating to patient transitions into the care home.
5. Codesign with relevant stakeholders a service specification for an integrated system response to safety incident reporting.

6. Map the codesigned service specification against existing systems to produce recommendations for implementation.
7. Investigate how the COVID-19 pandemic has influenced the management of safe transitions in care.

### Research questions

We will answer the following research questions during the study:

1. How do care homes currently respond to and report safety incidents for patients transitioning between hospital and care home?
2. What data do care homes currently collect on safety incidents relating to patients transitioning between hospital and care home, and what do the data tell us about the incidents that are reported?
3. What should an integrated system for learning from safety incidents that span health and social care organisations look like, and what would be required to implement this system?
4. To what extent has the COVID-19 pandemic influenced how transitions in care are managed safely?

## METHODS

This is a multimethod qualitative study, running from 4 January 2021 to 31 December 2022, consisting of two workstreams (see figure 1) that will run in parallel:

► Workstream 1: scoping of existing systems for safety incident reporting.
► Workstream 2: review of safety incident reporting and codesign of a service specification.

### Setting

For the purpose of this study a care home is defined as a residential care facility that provides temporary or permanent accommodation with nursing and/or personal care. Assisted living settings are excluded from the study. Workstream 1 will include data of relevance to all care homes across England. Workstream 2 will be situated within care home providers geographically separated (North East and South West England) representing approximately 30–50 care homes in each region, including nursing, residential and combined nursing/residential care. This will also include specialist care homes, for instance dementia care. Care home organisations will primarily be identified through existing networks and with the support of the National Institute for Health Research (NIHR) Clinical Research Network (CRN).

### Workstream 1

Workstream 1 will use desk-based approaches, defined as the collation of secondary data or data that can be collected without needing fieldwork, including searching online databases, the internet and organisation websites,[19] to investigate policies for incident reporting, technology used to incident report, and the types of data captured within incident reports. This will consist of three components: (1) structured telephone interviews with care home

# SafeST Study Flowchart

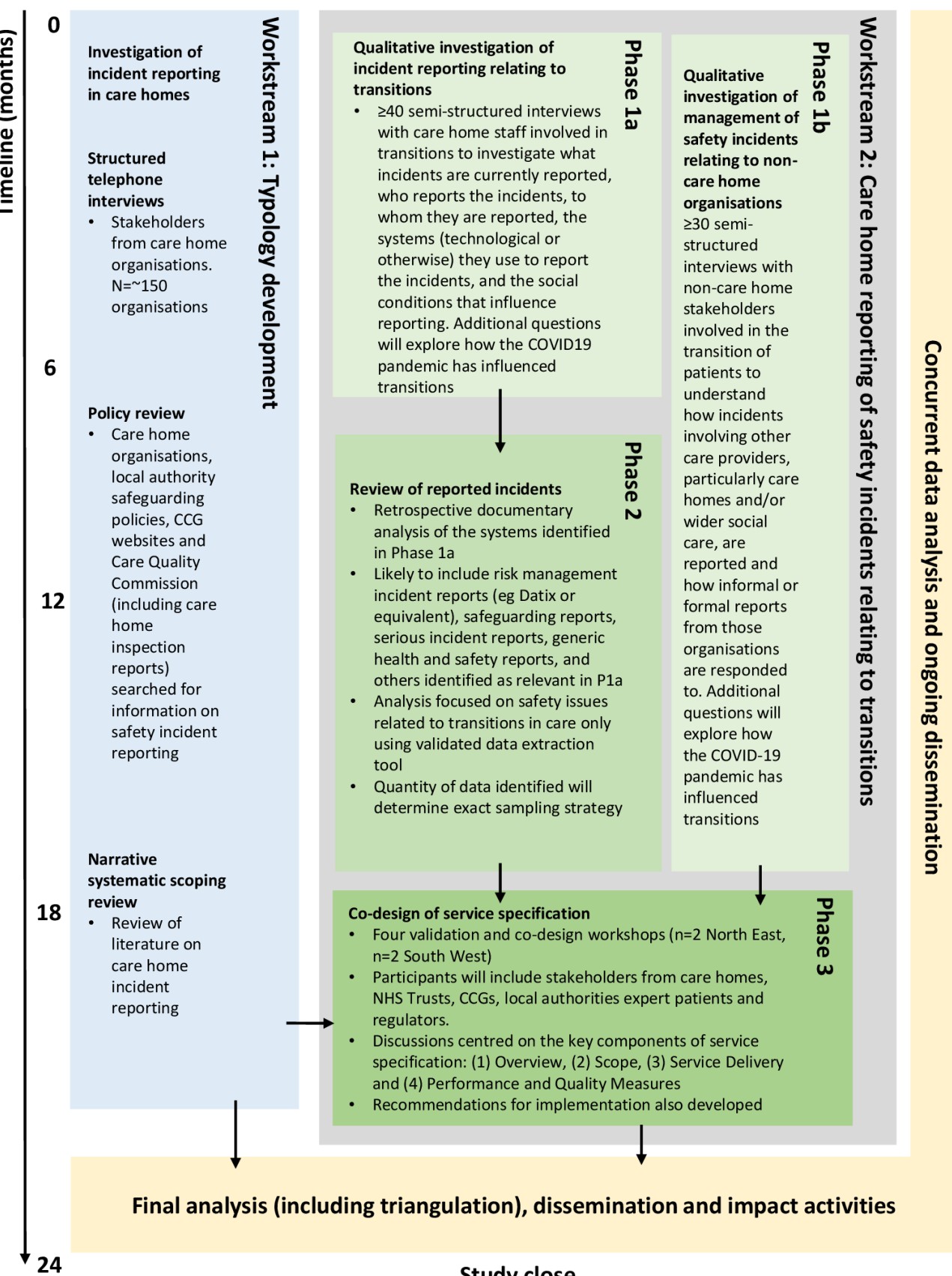

**Figure 1** Flow chart showing the configuration of the study's workstreams and phases.CCG; Clinical Commissioning Group, NHS; National Health Service.

**Table 1** Eligibility criteria and hierarchy of inclusion

| Eligibility criteria and hierarchy of inclusion | ▶ Published from 2000 onwards<br>▶ English language<br>▶ Empirical, peer-reviewed studies<br>▶ Populations must be in care homes (including residential and nursing homes)<br>▶ Issue or intervention must include incident reporting system(s), safety learning system(s), accident(s), and incident investigation system(s) |
| --- | --- |

managers, (2) a narrative scoping review of academic literature on existing incident reporting systems, and (3) a qualitative policy analysis of incident reporting policies related to care homes. This workstream will address study objectives 1 and 2. All three components will be used within the taxonomy development, but individually provide a full picture of incident reporting within care homes from research, practice and policy perspectives.

### Structured telephone interviews
#### Participants and sampling
Telephone interviews will be conducted with stakeholders who will be purposively sampled from 15 geographical regions in England based on the NIHR CRN footprint. This will provide a representation of different sized care home organisations (categorised relatively as small, medium and large), type of care provided (residential, nursing home or dual registration) and type of resident (general, dementia, learning disability, mixed). By sampling 10 care home organisations within each region, we anticipate a total sample of approximately 150 care home organisations, though some double counting may exist where large organisations span multiple regions.

#### Data collection
An interview guide has been developed and will be pilot tested prior to the start of data collection. Questions will cover topics related to incident reporting policy, approaches to incident reporting and management of incident reports. In addition to the structured interview data, data will also be collected on Care Quality Commission (CQC) ratings of care homes within the sampled organisations. The CQC ratings will be used retrospectively, in addition to the sampling criteria, to further determine representativeness of the sample. The ratings will be obtained from the CQC website (https://www.cqc.org.uk/) at the time of the telephone interview. Interviews will be voice recorded to allow the researcher to listen back, but will not be transcribed. Instead, data will be recorded in a spreadsheet based on the structured questions asked. The audio recording will be used to ensure an accurate interpretation is made of participants' responses, reducing the chance of researcher bias. Concurrent notetaking will also be conducted by the researcher, which together with the audio recordings has

been recognised as a suitable process for standardised open-ended interviews.[20]

### Review of literature
This review will follow the Preferred Reporting Items for Systematic Reviews and Meta-Analyses Protocols (PRISMA-P) statement.[21] The primary aim of the literature review will be to develop an understanding of incident reporting in care homes, with objectives focused on understanding policy, technology and types of data collected within incident reports.

#### Eligibility criteria
To determine eligibility, inclusion criteria will be applied, and a hierarchy of inclusion and exclusion has been developed to aid the sifting process (table 1). The hierarchy will be used to support the review team in highlighting at which point the paper was deemed to be ineligible for inclusion. At any point if the reviewer answers no, the paper will be deemed to be unsuitable for inclusion in the review with the reason recorded.

#### Search strategy
The SPIDER framework (*S*ample, *P*henomenon of *I*nterest, *D*esign, *E*valuation, *R*esearch type) will be used throughout the review. Specifically, the S, PI elements will inform the keywords, consisting of variations of "care home" (S) and "incident reporting" (PI), which have been adapted from previous relevant reviews.[22 23] To search for and identify any academic literature that is eligible the following platforms will be used:
- ▶ Cumulative Index to Nursing and Allied Health Literature (CINAHL), Medline and PsycINFO will be searched using the EBSCO platform.
- ▶ Embase and the Health Management Information Consortium (HMIC) will be searched using Ovid.
- ▶ Applied Social Sciences Index & Abstracts (ASSIA) and Nursing & Allied Health Database will be searched using ProQuest.
- ▶ Web of Science will be searched using Web of Science.
- ▶ Scopus being searched using Elsevier.

In addition to the formal academic database searches, grey literature will be searched using MedNar and OpenGrey. Handsearching will also take place of any included papers' reference lists to identify any potentially suitable papers that have not been identified via the database searches. A list of all search strings are reported in online supplemental material. The D, E and R elements of the framework will be used to inform data extraction.

#### Data management
All results from the academic literature databases will be retrieved. The first 100 hits from each of the grey literature databases will be retrieved as the first 100 hits are deemed to be sufficient.[24 25] All results will be downloaded into bibliographical software such as EndNote (Clarivate Analytics, V.X9). On transfer into bibliographical software, duplicate entries will be removed then the process of study selection will begin.

## Study selection

Papers will be assessed in line with the inclusion and exclusion criteria and study selection will be reported in line with the PRISMA flow chart.[26] Papers will initially be sifted based on their title and abstract and coded as potentially eligible or not eligible. One reviewer will sift all results and 20% of the results will be double sifted by another reviewer from the research team, with disagreements discussed to resolve conflict. Full texts of all potentially eligible papers will then be reviewed, with each paper independently assessed for eligibility by two reviewers and coded as eligible, not eligible or unsure. Any disagreements will be discussed until agreement is reached. If no agreement can be reached, a third reviewer will make the final decision. If a paper is considered not to be eligible, a reason for exclusion will be recorded based on the hierarchy of inclusion/exclusion. Finally, references and citations of all included papers will be compiled into a separate library, and the full study selection process will be conducted again and repeated until no new eligible papers are identified.

## Quality assessment

Critical Appraisals Skills Programme (CASP) tools[27] will be used to critically appraise any papers which are to be included and data extracted. The CASP tools have been chosen, over others, as there are various versions of the tools which fit with different methods, including Randomised Controlled Trials, Qualitative research, and Cohort studies. Quality of studies will not determine inclusion or exclusion.

## Data extraction and synthesis

Bespoke data extraction tools have been developed and will be independently piloted by two reviewers on six papers (three quantitative and three qualitative) and adapted iteratively to answer the review's research questions. For primary research, the authors, year of publication, country of study, aim of the research, study design, methods and study setting will be extracted. Following the review's research questions, data will also be extracted relating to the context of incident reporting, types of safety incident data reported, and the systems/technology used to facilitate incident reporting. Extraction of quantitative studies will also include the study measures, type of analysis, descriptive and inferential statistics. Extraction of qualitative studies will include the type of analysis, and summary of findings (including themes, categories, theories/models). Final data extraction will then be conducted by one reviewer, with a minimum of 10% independently double extracted. Disagreements will be resolved by discussion between the reviewers. If a consensus cannot be reached between the two reviewers, another member of the research team will be consulted.

Following data extraction, data will be synthesised using a three stages process: (1) free line-by-line coding of the findings, (2) organisation of these 'free codes' into related areas to construct 'descriptive' themes following the review's research questions, and (3) the development of 'analytical' themes.

## Policy review

### Sampling

Policies and guideline documents for incident reporting relating to care homes will be identified through internet searches, with a specific focus on providers (care homes), commissioners (Local Authorities, Clinical Commissioning Groups) and the main regulator (CQC). The search strategy will consist of keyword searches using Google search engine by combining each of these with 'incident reporting', for example: ("local authority" OR "care home" OR "Clinical Commissioning Group" OR "Care Quality Commission") AND "incident report". This will be supplemented by manual searching of the CQC website (https://www.cqc.org.uk/) for policies mentioning incident reporting and care homes, and all structured interview participants will be asked to share policies from their care home. To be included in the sample, policies must relate to safety incident reporting and either partially or wholly related to care homes. The first 100 hits will be retrieved, in line with recommended guidance for evidence reviews using Google.[24]

### Data collection

Data relating to the completion of incident reports, the process(es) for sharing incident reports, types of data collected, and technologies used will be extracted. Descriptive information, such as who the policy is aimed at and the date of the policy will also be extracted and a description of the wider context of the policy will also be created. Data will be extracted into a spreadsheet, and the data extraction process will be modified iteratively to allow for unexpected data to be identified. A portion (10%) of policies will be double coded at the start of the process, and major disagreements will be discussed between the coders prior to completing the remainder of the analysis.

## Workstream 2

Structured into three sequential phases (P1–P3), W2 will explore how care homes report safety incidents that relate to the wider health and social care system, and how non-care home stakeholders report incidents relating to other organisations. Workstream 2 will also examine how transition safety has been managed during the COVID-19 pandemic.

### Phase 1A: qualitative interviews with care home staff
#### Participants and sampling

Up to 40 semistructured interviews, which will allow for emerging and unexpected discussion, will be conducted with care home staff. Participants will be purposively sampled based on possible involvement in the transition of a patient, such as managers, nursing staff and healthcare assistants. It is anticipated that 40 interviews will provide sufficient information power[28] to meet the study objectives, and this will be reviewed regularly by the research team.

### Phase 1B: qualitative interviews with non-care home staff involved in transitions

#### Participants and sampling

Up to 30 semistructured interviews will be conducted with non-care home stakeholders involved in the transition of patients to understand how incidents involving other care providers, particularly care homes and/or wider social care, are reported and how informal or formal reports from those organisations are responded to. Participants will be purposively sampled to cover a spectrum of professionals, including social workers, nurses, care home linked general practitioners, occupational therapists and physiotherapists. We anticipate 30 participants to provide sufficient information power.[28]

#### Phases 1A and 1B data collection

Semistructured interviews will be more in-depth than in workstream 1 and will specifically focus on transitions both into and out of hospital. Questions will focus on what incidents are currently reported, who reports the incidents, to whom they are reported, the systems (technological or otherwise) they use to report the incidents, and the social conditions that influence reporting, including the 'work-as-done'[29] actions related to formal and informal incident reporting. Additional questions relating to COVID-19 will also explore with participants how the pandemic has influenced transitions between hospital and care home, including how safety behaviours and practices have (or have not) occurred during the pandemic, whether reporting of safety incidents has changed as a result of the pandemic, and the drivers for any such changes. Interviews will be conducted either face-to-face or remotely either by telephone or Microsoft Teams, digitally voice recorded, and transcribed verbatim using an external transcription company.

### Phase 2: retrospective documentary analysis

#### Data collection

A retrospective documentary analysis of the systems identified in phase 1a will be conducted, covering two financial years; 19/20 and 20/21. This time period will allow for seasonal variation and also prepandemic and intrapandemic variation. Data will include risk management incident reports (eg, Datix or equivalent), safeguarding reports, serious incident reports, generic health and safety reports, and others identified as relevant in phase 1a. The definition of a transition in care can be difficult to specify because effects of a poor transition can be long lasting.[30 31] For the purpose of this study, data will be included where there is specific mention of organisations or staff not part of the care home from which the data are obtained from. All data will be anonymised prior to transfer to the study team.

As the quantity of data will not be known until data collection begins, a decision will be made with advisory group input as to whether all data will be included. If not, a hybrid sampling model combining both purposive and random sampling will be used to ensure data is manageable

and representative. Purposive sampling will be conducted initially so as to ensure representation between relevant characteristics, such as the different technologies and systems in which the reports are contained, the type of harm, the person reporting and the incident and the level of harm. Where there are excessive data in any category to analyse within the study timeframe, we would then randomly sample within each category.

### Phase 3: service specification co-design

#### Participants and sampling

Codesign is recognised as a valid approach to sharing knowledge and expertise so long as power differentials are recognised and addressed to encourage inclusiveness.[32] Four validation and codesign workshops will be hosted (two representing North East England, two representing South West England), where results from phases 1 and 2 will be presented to care home and NHS staff involved in transition of patients. We will also invite commissioners of health and social care services, local authority safeguarding teams, regulators and patient and public involvement (PPI) representatives. Care home managers and nurses will be recruited, as will nursing staff from local NHS Trusts, specifically from wards or units that are most likely to discharge patients to nursing homes such as care of older people wards. We will endeavour to include participants from earlier phases, but this will depend on participant availability and provision of informed consent again. It is anticipated that each workshop will consist of between 12 and 15 participants, with participants in each workshop split into two groups for discussion, with one facilitator per group. Participants will only participate in one workshop each.

#### Data collection

During the workshops, a service specification will be codesigned for an integrated system response to safety incident reporting, and mapped against existing systems to produce recommendations. To codesign the service specification, the NHS Improvement[33] service specification template will be used, with discussions among participants centred on the key components of (1) overview, (2) scope, (3) service delivery and (4) performance and quality measures. A final discussion will also focus on recommendations for how to implement the service specification. During all discussions, we will employ various methods to encourage inclusion, based on previously published codesign research, and we will converge ideas between groups following each discussion.[34] Due to the COVID-19 pandemic, the workshops may be held online using specialist collaborative software (eg, Miro; www.miro.com) that allows participants to brainstorm, comment, draw and map plans asynchronously. Data from the four workshops will be collected independently, meaning that once data analysis is complete, outcomes from this phase may differ from those that participants have codesigned. To ensure any produced outcomes are

## Box 1 Lay summary of the study

When people move between hospital and a care home, it is quite common for something to go wrong with their care that does or could affect their safety. This is called a safety incident. Some examples include medicines being lost or delayed, or important documents containing mistakes or going missing. It is important to find out when and why safety incidents happen so that improvements can be made. Finding this out does not happen enough because care homes and hospitals sometimes have different priorities other than the person's care. They also have different understandings of what unsafe care means. Hospitals generally think of unsafe care as being a problem with the system that affects everyone, whereas care homes usually think of someone being unsafe because of problems with the care provided to just them.

Because of these different approaches, it can be difficult for care homes, hospitals, or even organisations that oversee them, to learn from safety incidents. As such, this study aims to understand how care homes and care home staff report safety incidents when a person moves between hospital and care home. Using this understanding, we aim to work with care homes and hospitals to jointly design a better way of reporting and learning from safety incidents.

The study will be split into two parts that run alongside each other. During the first part (workstream 1), we will review how care homes respond to safety incidents. This will include looking at what policies exist, what technology is used and how reports are captured. This review will be desk-based, combining internet searches, telephone interviews and academic papers. From this, we will create categories of the different systems being used to capture safety incidents. During the second part (workstream 2), we will work with two care home organisations, one in North East and one in South West England. Each will contain around 30–50 care homes, and will cover all different types of care homes (eg, nursing, residential, dementia). We will begin by speaking with up to 40 care home staff to find out how they report incidents. Separately, we will also speak with up to 30 staff from other organisations, such as hospitals, who are involved in people moving between hospital and care home. We will then review the information that the care homes hold, using a method that we have developed and used previously. Using what we learn, we will jointly design a new system for learning from safety incidents by hosting four workshops and inviting people who will have suitable experience. This will include care home mangers and nurses, clinical staff from National Health Service trusts, commissioners of health and social care services, local authority safeguarding teams, regulators and patient and public involvement representatives. This new system will be compared with existing systems and recommendations will be made for how it can be put into practice.

valid, they will be shared using Miro with all workshop participants for final discussion and verification.

### Data analysis

Data from workstream 1 will be used to develop a taxonomy of approaches to reporting incidents in care homes using a standardised approach to taxonomy development,[35] aligned with a combination of objective and subjective ending conditions. Objective conditions will include no new dimensions being added or amended in the final iteration, there is no duplication across dimensions, and each dimension has a minimum of one characteristic. Subjective conditions include the taxonomy being concise, robust, comprehensive, extendible and explanatory.

Framework analysis will be used concurrently to analyse data from across the whole study, while also allowing for the emergence and identification of new themes.[36] The framework will be based around the study objectives, with an additional analysis heuristic of systemic risk factors, specifically latent conditions and active failures that contribute to organisational learning, as well as the proximal and distal factors. This heuristic will provide evidence for the wider system-level factors that contribute to safety incidents that are reported in care homes. The exception to this approach is in phase 2, where content analysis will be used to analyse documents.[37] While documentary analysis has the benefit of not being influenced by the data collection method, other biases are likely to exist,[38] particularly related to under-reporting of incidents. Documentary analysis will focus specifically on safety issues related to transitions in care, using a prevalidated method and data extraction tool.[39]

Analysis will include triangulation, particularly examining conflicts or disagreements within the data. A convergent coding matrix[40] will be used to facilitate the triangulation of data from multiple sources. Following completion of both workstreams, the results of workstream 2 will be cross-referenced to determine the transferability of the findings across the taxonomy developed in workstream 1.

### Ethics and dissemination

The study has received ethical approval from Northumbria University (ref: 120/2450) and has received Health Research Authority (HRA) approval (ref: 20/HRA/5272).

Study findings will be disseminated via numerous avenues. A study website, which is publicly accessible, will be launched (https://research.northumbria.ac.uk/SafeST) and will provide an overview of the study design, study findings and completed dissemination. Findings will also be disseminated directly to participants where they have been requested, will be published in peer-reviewed journal articles, and disseminated at academic and practice conferences.

### Patient and public involvement

PPI is embedded in the study. As a coapplicant, MS has been directly involved in helping to design the PPI strategy, including identifying the need to involve additional patient, carer and public representatives in the wider advisory group, with appropriate support, to ensure that their voice is heard. MS helped to facilitate a presentation to the North Tyneside Community and Health Care Forum, specifically in relation to the appropriateness and value of the research question, how best to raise awareness of the study while it is underway among patients, carers and the wider public, and how best to disseminate the findings to patients, carers and the wider public. The Forum will provide advice throughout the study as a PPI advisory group, and there will be PPI representation

on the study advisory group. We will engage with these groups on an ongoing basis to determine how best to disseminate the findings to patients and the public. Box 1 presents a lay summary of the research.

## DISCUSSION

The novelty of this research is threefold. First, this will be the first study examining how safety incidents are reported across the care home sector in relation to transitions in care using data generated within the care home sector. In doing so, the study will produce knowledge of relevance to both health and social care services that can improve transitions in care. Also by developing a better understanding of incident reporting within this setting, the study will be able to make recommendations for improving incident reporting, which can improve organisational learning and therefore the quality of care of older people. By investigating the impact of the COVID-19 pandemic on the safety of care transitions, the study will provide additional vital insight and learning. Second, by codesigning a service specification, we aim to begin to address the under-reporting of incidents within this setting by defining the service standards expected from organisations involved in patient transitions, for instance hospital and care home providers, as well as commissioners and regulators. Third, the developed taxonomy of approaches to incident reporting systems within the care home sector will provide a foundation for future research within this setting. The taxonomy will also provide the foundations required for implementing changes within the sector that are required for making improvements to the care and support of older people transitioning between hospital and care home.

**Acknowledgements** The authors would like to acknowledge the support of the National Institute for Health Research Clinical Research Network. The views expressed are those of the authors and not necessarily those of the NIHR or the Department of Health and Social Care. We would also like to thank members of the North Tyneside Community and Health Care Forum for providing valuable input into the design of the study.

**Contributors** JS, KBr, PD, LY-M, MS and JW contributed to the conception and design of the study and are the grant holders. KBy and SM will lead data collection and analyses with input from all authors. JS wrote the manuscript with all authors contributing to the drafting and revision of the manuscript, and all authors approved the final version.

**Funding** This work is supported by The Dunhill Medical Trust, grant number RPGF2006\226.

**Competing interests** None declared.

**Patient consent for publication** Not applicable.

**Provenance and peer review** Not commissioned; externally peer reviewed.

**ORCID iD**
Jason Scott http://orcid.org/0000-0001-7031-2171

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
