## [Reviewer comments · BMJ Open]

ARTICLE DETAILS

TITLE (PROVISIONAL)	Residents transitioning between hospital and care homes: Protocol for co-designing a systems-level response to safety issues (SafeST study)
AUTHORS	Scott, Jason; Brittain, Katie; Byrnes, Kate; Dawson, Pam; Mulrine, Stephanie; Spencer, Michele; Waring, Justin; Young-Murphy, Lesley

VERSION 1 – REVIEW

REVIEWER	Abebe, Ephrem Purdue University College of Pharmacy, College of Pharmacy, Purdue University, West Lafayette, IN, USA
REVIEW RETURNED	24-Aug-2021

GENERAL COMMENTS	Introduction Well written and provides sufficient context for remainder of manuscript. One suggestion I have is to define what you mean by “care homes”. There are several terms used in different countries/contexts to describe the care setting you are referring to. I think the reader will be best served by having a brief definition, what they constitute, and what type of care is delivered in these settings, etc. For example, readers in some countries may be familiar with terms like skilled nursing facility, assisted living but not such much about care homes. Aims and objectives 1. What do you mean by desk-based approaches? Referring to review of existing documents, policies? Methods Workstream 1 Structured telephone interviews -The connection of Care Quality Commission (CQC) ratings to interview data is not clear. What is the purpose of gathering these ratings? Review of literature -During initial screening for titles and abstract (under study selection), can you describe how you will resolve any conflicts with the two reviewers on the 20% articles that are double screened? Policy Review -Could you be missing policy documents that are not uploaded to the internet? For example, could your focus organizations make these documents available on their local network but not on public facing websites? Workstream 2 Phase 3: Service specification co-design
--

-For your co-design workshops, how many participants to do anticipate enrolling per session?
 -You report that service specification will be the expected output of co-design workshops. Will your team be doing any evaluation (e.g, formative) co-design output?

VERSION 1 – AUTHOR RESPONSE

Reviewer	Comment	Response	Location in manuscript
1	Introduction. Well written and provides sufficient context for remainder of manuscript. One suggestion I have is to define what you mean by “care homes”. There are several terms used in different countries/contexts to describe the care setting you are referring to. I think the reader will be best served by having a brief definition, what they constitute, and what type of care is delivered in these settings, etc. For example, readers in some countries may be familiar with terms like skilled nursing facility, assisted living but not such much about care homes.	Thank you for this suggestion and we completely agree that it would help given the complexity of care home nomenclature. We have now included a definition of care homes, which has been positioned at the start of the Methods > Setting section of the manuscript. We have also made a minor amendment to the introduction to now mention the Learn from patient safety events (LFPSE) service. This service is currently being rolled out to replace the National Reporting and Learning System, and has only recently been announced.	P4, lines 151-152 P3, lines 82-83
1	Aims and objectives. What do you mean by desk-based approaches? Referring to review of existing documents, policies?	Desk-based research refers to the collation of secondary data or data that can be collected without needing fieldwork, including searching on line databases, the internet and organisation websites (Hague, 2006). We have now included this definition in Methods > Workstream 1 where desk-based approaches are first discussed (other than the aims/objectives).	P4, lines 162-164
1	Methods, Workstream 1, Structured telephone interviews. The connection of Care Quality Commission (CQC) ratings to interview data is not clear. What is the purpose of gathering these ratings?	The CQC ratings will be used retrospectively, in addition to the sampling criteria, to further determine representativeness of the sample. We have included this in the data collection section.	P5, lines 190-191
1	Methods, Workstream 1, review of literature. During initial screening for titles and abstract (under study selection), can you describe	Thank you for spotting this, we have now amended to say conflict will be resolved through discussion.	P6, lines 246-247

	how you will resolve any conflicts with the two reviewers on the 20% articles that are double screened?		
1	Methods, Workstream 1, Policy Review. Could you be missing policy documents that are not uploaded to the internet? For example, could your focus organizations make these documents available on their local network but not on public facing websites?	Yes, it is possible that policies may be missed using this approach. We have tried to mitigate against this scenario by asking all participants from the structured telephone interviews to share their policies, regardless of whether they are available on websites or not. We have amended the text of this section to make it clearer that all interview participants (and intended sample of 150 care homes) will be asked to share policies.	P7, lines 291-293
1	Workstream 2, Phase 3. Service specification co-design -For your co-design workshops, how many participants to do anticipate enrolling per session?	Thank you for spotting this omission. We have now included an anticipated sample size (12-15 participants per workshop) in this section.	P9, lines 382-383
1	Workstream 2, Phase 3. You report that service specification will be the expected output of co-design workshops. Will your team be doing any evaluation (e.g, formative) co-design output?	This is an excellent point to raise, thank you. We have now explained in the data collection section that each workshop will be independent, and a final stage of data collection will occur asynchronously using Miro, which will form participant verification of the final outcomes.	P9, lines 396-399.

VERSION 2 – REVIEW

REVIEWER	Abebe, Ephrem Purdue University College of Pharmacy, College of Pharmacy, Purdue University, West Lafayette, IN, USA
REVIEW RETURNED	04-Nov-2021

GENERAL COMMENTS	Thank you to authors for providing clarification. All my comments have been addressed in this revised version. One concern I have is that authors wrote they expect 12-15 participants in their co-design workshop. This number seems a bit high for a co-design methodology and I wonder if the researchers will have enough time to move through stages of the co-design process (while ensuring meaningful input from all participants). It is also not clear if participants will participate in just one co-design workshop or multiple rounds of co-design workshops.
---

VERSION 2 – AUTHOR RESPONSE

Reviewer	Comment	Response	Location in manuscript
1	One concern I have is that authors wrote they expect 12-15 participants in their co-design workshop. This	We have provided further clarification that participants will be split into two groups, with a single facilitator each and thus having more manageable group sizes. This is an approach	P9, lines 379 – 380 P9, lines 391-392

	number seems a bit high for a co-design methodology and I wonder if the researchers will have enough time to move through stages of the co-design process (while ensuring meaningful input from all participants).	we have used successfully in previous co-design work, which is cited in the manuscript.	
1	It is also not clear if participants will participate in just one co-design workshop or multiple rounds of co-design workshops.	We have likewise provided further clarification that participants will take part in one workshop each. This was also included in the data collection section that stated “Data from the four workshops will be collected independently”, but the extra wording now makes it clearer.	P9, lines 379 – 381